# OpenReview forum: "Memory Caching: RNNs with Growing Memory"
_ICLR.cc/2026/Conference — Submitted to ICLR 2026_

### Official Review · Reviewer_3LNG · 2025-10-22

**Soundness:** 3
**Presentation:** 3
**Contribution:** 2
**Rating:** 4
**Confidence:** 5

**Summary:**

This paper proposes Memory Caching (MC), a technique that enhances the memory capacity of RNN by caching checkpoints of memory states, thus allowing memory to grow with the sequence length. The MC framework aims to strike a balance between the fixed memory size of traditional RNNs $(O(L))$ and the growing memory complexity of Transformers $(O(L^2))$. The paper introduces several MC variants, including Gated Residual Memory (GRM), Memory Soup, and Sparse Selective Caching (SSC). Experiments on language modeling and retrieval tasks show that MC improves RNNs’ retrieval performance and can close the gap with Transformer-based models, though Transformers still outperform in recall-intensive tasks.

**Strengths:**

1. The paper effectively addresses the bottleneck of fixed-size memory in RNNs by proposing a mechanism that allows memory to grow with the sequence length, providing a flexible middle ground between traditional RNNs and Transformer models.
2. The MC framework, especially with Gated Residual Memory and Sparse Selective Caching, significantly enhances the retrieval ability of RNNs, improving performance on tasks where sequence history plays a crucial role.

**Weaknesses:**

1. The proposed GRM still leads to quadratic complexity in the worst case when applied to long sequences, making it no better than vanilla attention in terms of computational complexity. If SWA (Sliding Window Attention) being treated as an RNN, SWA+GRM essentially reduces to vanilla attention, undermining the claim of significant computational efficiency improvements over attention-based approaches like Transformers.
2. Despite the enhancement in memory efficiency and retrieval performance, the results still show that Transformers outperform MC-enhanced RNNs on recall-intensive tasks. This suggests that simply expanding memory is not enough to fully match the expressive power of Transformer models, particularly in tasks that require broader context retention.

**Questions:**

1. Given that the GRM design still results in quadratic complexity for long sequences, what specific advantage does this design offer over standard attention-based methods that already deal with growing memory in sequences? Is the claimed advantage over Transformers primarily empirical rather than computational?
2. While expanding the memory size through caching improves retrieval performance, why does it not fully align with Transformer models in recall-intensive tasks? Could this be due to inherent limitations in RNN architectures when it comes to capturing long-term dependencies, even with larger memory?

---

> ### Author Response · Authors · 2025-12-04
>
> Thank you for your time and valuable comments. We really appreciate it.
>
> > The proposed GRM still leads to quadratic complexity in the worst case when applied to long sequences, making it no better than vanilla attention in terms of computational complexity. If SWA (Sliding Window Attention) being treated as an RNN, SWA+GRM essentially reduces to vanilla attention, undermining the claim of significant computational efficiency improvements over attention-based approaches like Transformers.
>
> **Response:** We respectfully disagree and want to bring to your consideration that the complexity of GRM relies on the choice of segmentation. With segmenting the sequence into subseqeunces with exponentially growing length the complexity of GRM is $N \log N$. In the worst case, when we use the same segment size for all segments, the complexity of GRM still relies on the segment length. That is, considering the case of having sequence length of 32K: (1) the cost of vanila attention is in $O((32K)^2)$ operations. (2) The cost of GRM with segment size of 128 is in $O((32K/128)^2)$ or equivalently $O((256)^2)$. Accordingly, GRM can be about 16000 times less costly than a sotmax attention. With increasing the segment size, this theoretical cost enhancement can become even larger. Therefore, we do not see this as a weakness of our approach, but as its strength that can interpolate between the complexity of softmax attention and recurrent architectures.
>
>
>
>
> > Despite the enhancement in memory efficiency and retrieval performance, the results still show that Transformers outperform MC-enhanced RNNs on recall-intensive tasks. This suggests that simply expanding memory is not enough to fully match the expressive power of Transformer models, particularly in tasks that require broader context retention.
>
> **Response:** Please note that Transformers have perfect memory, meaning that it stores all past keys and values with no compression. Therefore, when there is a recall-intensive task, it is not often the case that an alternative architecture could outperform Transformers. The similar patterns have also be observed in other recurrent architectures. Memory Caching, however, closes this gap of recurrent architectures and Transformers and improves the performance of recurrent base models.

---

### Official Review · Reviewer_a1Gd · 2025-10-23

**Soundness:** 3
**Presentation:** 3
**Contribution:** 3
**Rating:** 4
**Confidence:** 4

**Summary:**

The authors introduce Memory Caching (MC), a technique designed to enhance Recurrent Neural Networks (RNNs) by allowing their effective memory capacity to grow with the input sequence length, similar to Transformers. The technique involves caching checkpoints of the RNNs' memory states (hidden states) along the sequence. This effectively allows the model's memory to expand as the sequence grows. The experimental results demonstrate that MC (and the multiple proposed variants) are effective for enhancing recurrent models on multiple recall-intensive tasks and long-context understanding settings relative to existing state-of-the-art recurrent models.

**Strengths:**

The results are well presented and the method is intuitive to follow. Experimental results show cases where improvements are significant.

**Weaknesses:**

- The experiments are missing a rather important training and inference efficiency comparison of using the cached memories. In general, if the caching is done at a fixed length (suppose $L$), then the memory will grow linearly with the sequence length, potentially limiting the ability of the memory to train on longer contexts as well as conduct significantly longer inference compared to alternative linear models, which normally operate in $O(1)$ memory at inference time.
- The main area where results improve is in retrieval-based tasks, which is rather intuitive since there is an additional cache of prior information that is being used and incorporated within the model.
- I believe rather than comparing against Samba, Hymba would be a more fair comparison as the information mixing happens between parallel branches in the same layer, thus it acts closer to a memory cache unlike Samba where the interleaving occurs between layers.

[1] Hymba: A Hybrid-head Architecture for Small Language Models

**Questions:**

- Given the scale of the models, it might be more convincing to incorporate the method to additional linear RNN or linear attention based methods such as Mamba/Mamba2 and GLA/GSA/DeltaNet/GatedDeltaNet.

- As a follow up to the first weakness, I would suggest the authors look into verifying whether the method would be useful as a post-tuning augmentation of already pre-trained models. Given the additional memory requirements, I have hesitations about whether or not the proposed method would be feasible for pre-training in most settings but for additional calibration or tuning of models for longer contexts (which I believe is the primary motivation of the authors) there may be greater space to operate. Have the authors attempted this?

---

> ### Author Response · Authors · 2025-12-04
>
> Thank you for your time and valuable comments. We really appreciate it.
>
>
> > The experiments are missing a rather important training and inference efficiency comparison of using the cached memories.
>
> **Response:** Following your suggestion, we have provided new results on the efficiency of our approach. Please note that memory caching not only improves the performance, but it also unlocks the sequence parallelization in recurrent models, resulting in much faster training.
>
> | Model | Training Throughput (8K) $\approx10^{3}T/s$ |
> | :--- | :---: |
> | Transformer | 48 |
> | Mamba/DeltaNet/Titans | 33 / 39 / 37 |
> | **Memory Caching** | **49 / 48 / 46** |
>
>
> > The main area where results improve is in retrieval-based tasks, which is rather intuitive since there is an additional cache of prior information that is being used and incorporated within the model.
>
> **Response:** Please note that the improvement MC shows even in language modeling tasks is considered significant and is already more than the rerported improvement in peer papers. Also, please note that, we overcome an important drawback of RNNs in this paper, which has been fixed-size state.
>
>
>
> > I believe rather than comparing against Samba, Hymba would be a more fair comparison as the information mixing happens between parallel branches in the same layer, thus it acts closer to a memory cache unlike Samba where the interleaving occurs between layers.
>
> **Response:** Thank you for your suggestion. Following your suggestion, we have added Samba as the baseline to our evaluations. Our variants of memory caching outperforms Samba in language modeling and common-sense reasoning tasks. Please note that: both Samba and Hymba are hybrid models and our memory caching method is still a recurrence-based architecture. Therefore, in our initial experiments, we only compared with state-of-the-art recurrent architectures and show that Memory Caching on top of those methods can enhances the performance of the baselines. Note that, still our memory caching technique can be applied to the recurrent blocks of hybrid architectures such as Samba/Hymba, further enhancing their performance.
>
>
> > Given the scale of the models, it might be more convincing to incorporate the method to additional linear RNN or linear attention based methods such as Mamba/Mamba2 and GLA/GSA/DeltaNet/GatedDeltaNet.
>
> **Response:** Thank you for your suggestion. Please note that similar to other peer studies, we have shown the effectiveness of memory caching on two different recurrent architectures (with different learning rules). Indeed there is always a room for adding more baselines and it can be an important future direction. Our current experiments, at least are showing improvement over two SOTA models both as a pre-training method or a post-tuning process.
>
>
> > As a follow up to the first weakness, I would suggest the authors look into verifying whether the method would be useful as a post-tuning augmentation of already pre-trained models. Given the additional memory requirements, I have hesitations about whether or not the proposed method would be feasible for pre-training in most settings but for additional calibration or tuning of models for longer contexts (which I believe is the primary motivation of the authors) there may be greater space to operate. Have the authors attempted this?
>
> **Response:** Thank you for this great suggestion. Following your suggestion, we also evaluate the performance of memory caching as a post-tuning process and the results are as follows:
>
> | Model | Language Modeling (the more negative the better) | Common-sense reasoning | RULER
> |---|---|---|---|
> | + **Memory Caching** |  -0.94  | + 0.38 | +6.3 |
>
> Memory caching can significantly enhance the length extrapolation capability of the base model, making it an ad-hoc method for enhancing the capability of recurrent models even after training.

---

### Official Review · Reviewer_9kEz · 2025-11-01

**Soundness:** 3
**Presentation:** 3
**Contribution:** 3
**Rating:** 6
**Confidence:** 4

**Summary:**

This paper introduces Memory Caching (MC), a simple yet effective technique that enhances RNN models that allows memory states grow with sequence length. Experiments demonstrate the effectiveness of proposed method on language modeling, long context understanding and efficiency.

**Strengths:**

1. Designing a trade-off architecture between the computational complexity of RNNs and transformers is a valuable research area.
2. The proposed method is simple yet effective, and appears to be applicable to various RNN architectures.
3. The paper is clearly structured and includes sufficient experiments.

**Weaknesses:**

1. The paper seems to lack analysis on the impact of the number of input sequence segments on model performance. This could help us identify the trade-off between the complexity of RNN and transformer architectures and explore better memory caching lengths when the model performs sequence modeling.
2. A characteristic of RNN model architectures is their potential to generalize to longer sequences, but the paper does not seem to focus on the performance of the proposed method in terms of length extrapolation.

**Questions:**

1. Why do the proposed method and various other linear RNN models shown in Table 1 significantly outperform Transformers++? Since Transformers++ uses a key-value cache to store all the historical information, shouldn't its performance be the best? I referred to the results of some RNN-related papers [1][2][3], and although RNNs show high efficiency with linear complexity, their performance still seems to lag significantly behind the Transformer.
2. Gated Residual Memory (GRM) seems to outperform other memory caching method variants in all tasks. Can you explain this situation?

[1] Songlin Yang, et al. Gated Linear Attention Transformers with Hardware-Efficient Training. ICML, 2024.
[2] Songlin Yang, et al. Parallelizing Linear Transformers with the Delta Rule over Sequence Length. NeurIPS, 2024.
[3] Jiaxi Hu, et al. Improving Bilinear RNNs with Closed-loop Control. NeurIPS, 2025.

---

> ### Author Response · Authors · 2025-12-04
>
> We thank the reviewer for their time and valuable comments, really appreciate it. We respond to your comments below:
>
>
> > The paper seems to lack analysis on the impact of the number of input sequence segments on model performance.
>
> **Response:** Thank you for your suggestion. Following your suggestion and to better understand the effect of number of segments and their length on the model performance, we have added a new ablation where we vary the length of segments. The results are as follows (perplexity in language modeling):
>
>
> | 8 | 32 | 64 | 128 |
> |-|-|-|-|
> | 16.38 | 14.91 | 15.26 | 17.09 |
>
>
> > A characteristic of RNN model architectures is their potential to generalize to longer sequences, but the paper does not seem to focus on the performance of the proposed method in terms of length extrapolation.
>
> **Response:** We want to kindly bring to your consideration that:
> - RNNs are well-known to suffer from length extrapolation (due to fixed state size) and several studies have shown that in tasks such as NIAH, they fall short compare to Transformers, when both are trained using the same sequence lenght.
> - Please note that our experiments on NIAH tasks (Table 2) are in fact experiments on the length extrapolation of the models.
>
>
> > Why do the proposed method and various other linear RNN models shown in Table 1 significantly outperform Transformers++?
>
> **Response:** Thank you for this great question. Please note that the main goal of Table 1 is to fairly compare the power of seqeunce models backbones in language modeling tasks in a **controlled** manner. We acknowledge that there are variants of Transformers with additional components and normalization terms that can show better performance and outperform models; however, such variants uses normalization terms that are not commonly used in recurrent models. Our initial experiments show that such modifications also can enhance the performance of recurrent models; however, in our setup of Table 1, we use the same backbone for all the architectures and only replace the sequence model block everytime with one of the baselines or our own models.
>
> Another important point about the common-sense reasoning tasks used in Table 1 is that, they require short context understanding and so the exponential growing and direct access to past tokens, as in Transformers, might not help significantly. The similar observation also has been reported in Gated DeltaNet paper.
>
>
> > Gated Residual Memory (GRM) seems to outperform other memory caching method variants in all tasks. Can you explain this situation?
>
> **Response:** Thank you for this great question. Please note that comparing to our SSC variant, these results are fully expected as SSC only improves the efficiency of GRM and chooses a subset of cached memories. Comparing to Memory Soup variant also, both GRM and Memory Soup can be seen as RNNs, where the state of the memory for each token is different. From this viewpoint, GRM has a better memory selection as the entire memory with its deep structure is chosen; in memory soup, however, this process happens at the layers of the memories, potentially causing suboptimal memory fusion.  Following your suggestion, we will add more detailed discussion about the connection of variatns in the final version.

---

### Meta-Review · Area_Chair_59EY · 2026-01-08

**Summary:**

The paper proposes Memory Caching, a technique designed to bridge the gap between RNNs and Transformers. The primary concern driving the rejection is that the proposed Memory Caching technique occupies an ambiguous "middle ground" that compromises the core advantages of both RNNs and Transformers without offering a distinct enough benefit.

**Reviewer Concerns:**

The primary concern is the theoretical positioning of the method.

Whether this method provides sufficient value over sophisticated hybrid architectures remains relevant.

Despite the caching mechanism, the method still underperforms Transformers on recall-intensive tasks. As noted by the reviewers, if a task requires high-fidelity recall, users will likely opt for a Transformer.

**Reviewer Scores:**

The paper was reviewed by three reviewers who assigned ratings of 6, 4, and 4. While the reviewers acknowledged the intuitive nature of the approach and the clear presentation, there is a consensus that the paper falls short of the acceptance.

---

### Decision · Program_Chairs · 2026-01-26

Reject